# The Role of the Tumor Microenvironment (TME) in Advancing Cancer Therapies: Immune System Interactions, Tumor-Infiltrating Lymphocytes (TILs), and the Role of Exosomes and Inflammasomes

**DOI:** 10.3390/ijms26062716

**Published:** 2025-03-18

**Authors:** Atef M. Erasha, Hanem EL-Gendy, Ahmed S. Aly, Marisol Fernández-Ortiz, Ramy K. A. Sayed

**Affiliations:** 1Department of Anatomy and Embryology, Faculty of Veterinary Medicine, Sadat City University, Sadat City 32897, Egypt; atef.areisha@vet.usc.edu.eg; 2Department of Pharmacology, Faculty of Veterinary Medicine, Sadat City University, Sadat City 32897, Egypt; hanem.elgendy@vet.usc.edu.eg; 3Department of Animal Production, Faculty of Agriculture, Ain Shams University, Cairo 11241, Egypt; a_yousef129@agr.asu.edu.eg; 4Greehey Children’s Cancer Research Institute, University of Texas Health Science Center San Antonio, San Antonio, TX 78229, USA; 5Department of Anatomy and Embryology, Faculty of Veterinary Medicine, Sohag University, Sohag 82524, Egypt; ramy.kamal@vet.sohag.edu.eg

**Keywords:** cancer therapy, exosomes, immune cells, inflammasomes, tumor infiltrating lymphocytes (TILs), tumor microenvironment (TME)

## Abstract

Understanding how different contributors within the tumor microenvironment (TME) function and communicate is essential for effective cancer detection and treatment. The TME encompasses all the surroundings of a tumor such as blood vessels, fibroblasts, immune cells, signaling molecules, exosomes, and the extracellular matrix (ECM). Subsequently, effective cancer therapy relies on addressing TME alterations, known drivers of tumor progression, immune evasion, and metastasis. Immune cells and other cell types act differently under cancerous conditions, either driving or hindering cancer progression. For instance, tumor-infiltrating lymphocytes (TILs) include lymphocytes of B and T cell types that can invade malignancies, bringing in and enhancing the ability of immune system to recognize and destroy cancer cells. Therefore, TILs display a promising approach to tackling the TME alterations and have the capability to significantly hinder cancer progression. Similarly, exosomes and inflammasomes exhibit a dual effect, resulting in either tumor progression or inhibition depending on the origin of exosomes, type of inflammasome and tumor. This review will explore how cells function in the presence of a tumor, the communication between cancer cells and immune cells, and the role of TILs, exosomes and inflammasomes within the TME. The efforts in this review are aimed at garnering interest in safer and durable therapies for cancer, in addition to providing a promising avenue for advancing cancer therapy and consequently improving survival rates.

## 1. Introduction

According to the National Cancer Institute (NCI) and the Centers for Disease Control and Prevention (CDC), cancer represents a significant global health problem, encompassing over 100 different types that may originate in specific tissues and spread throughout the body. It is among the leading causes of morbidity and death worldwide [1,2]. Scientists developed a variety of cancer therapies, such as chemotherapy, radiotherapy, surgical therapy, and immunotherapy, which are currently available. These treatments can harm healthy cells and may cause other adverse effects. Consequently, scientists are investigating innovative approaches to target and eliminate tumor cells while selectively minimizing side effects.

The immune system has a crucial defensive role against tumors and can inhibit their progression. Initially, immunotherapy exhibited limited effectiveness, causing a lack of interest in cancer immunotherapy. However, later advances revealed that immune treatments may result in durable responses in a specific subgroup of patients. For example, interleukin-2 (IL-2) and adoptive transfer of autologous tumor-infiltrating lymphocytes (TILs) were found to increase the chances of long-term survival, even in patients with a poor prognosis [3]. Additionally, TILs hold significant promise as a form of cancer therapy. These lymphocytes can recognize and destroy cancer cells (tumoricidal activity) and halt tumor progression (tumorostatic role). As a result, the level of these lymphocytes can be used as an indicator of an individual’s immune response to tumors [4]. Identifying and quantifying TILs is possible through quantifying CD3 markers or employing techniques like gene expression microarray analysis or RNA sequencing. The presence of tumor-infiltrating T lymphocytes can be increased when anti-programmed cell death protein 1 (anti-PD1) therapy is utilized by using drugs like Pembrolizumab, Nivolumab, and Cemiplimab [5,6,7].

Cancer cells can employ different mechanisms to avoid their destruction by the immune system. They can stimulate immune cell suppressors, such as regulatory T cells (Tregs) and myeloid-derived suppressor cells, which inhibit the proliferation of T-cells and can even suppress the TILs [8,9]. Another mechanism includes the upregulation of programmed death ligand-1 (PD-L1), also called B7-H1, which can bind to the programmed death-1 receptor (PD-1) on the surface of T-cells, impairing their activity. In addition, tumor cells enhance the levels of reactive oxygen species and release substances like transforming growth factor-β (TGF-β), IL-10, exosomes, and nitric oxide, all of which display a direct and positive relationship with immunosuppression [10,11]. The heterogeneity of the tumor microenvironment (TME), composed of immune cells, stromal cells, blood vessels, extracellular matrix (ECM), and exosomes, contributes to this immune evasion, thereby impairing the efficacy of the immunotherapy [12,13,14]. The cellular components of the TME recruit and secrete protective cytokines, leading to treatment resistance. Non-cellular components of the TME, like ECM, as well as conditions that include hypoxia, high acidity and lactate, and the depletion of glucose and amino acids, act as a physical barrier to immunotherapy and impair the normal metabolism of lymphocytes. Exosomes produced by cancer cells in the TME facilitate inflammation, angiogenesis, tumor progression, and metastasis.

Exosomes are microscopic vesicles enclosed by a phospholipid bilayer, and they can be derived or secreted by most living cells. These membranous structures can transport various active biomolecules from one cell to another or from hosts to recipients. Recently, exosomes have emerged as potential vehicles for drug delivery, including both biological and non-biological medications. In this regard, exosomes offer several advantages over liposomes and nanoparticles, including low clearance rates, higher bioavailability (they can easily pass through biological barriers including the blood–brain barrier, placental barrier, and intestinal barrier), minimal immunogenicity, low cumulative toxicity in normal tissues, and the capacity to specifically deliver anti-cancer drugs to cancer cells by utilizing ligand–receptor interaction or endocytosis, thereby addressing drug resistance caused by P-glycoprotein or other multidrug resistance-associated problems [15,16].

Inflammasome-induced inflammation has been associated with the onset and progression of cancer. By activating caspase-1, inflammasomes initiate inflammatory responses that result in the release of pro-inflammatory cytokines such as IL-1β and IL-18 [17,18,19]. These cytokines impact tumor development by altering the TME, regulating immune responses, and facilitating cancer cell survival and metastasis. Inflammasomes can be activated in various cell subpopulations within the TME, including tumor cells, tumor-associated macrophages (TAMs), tumor-associated fibroblasts, and marrow-derived suppressive cells [20,21,22,23]. Recent studies highlight the dual role of inflammasomes in the TME, where they can either promote or inhibit tumor progression, depending on the specific inflammasome type and tumor context [24].

Exosomes play a crucial role as mediators of communication between tumor cells, immune cells, and other components of TME. Their influence on inflammasome activation varies, either enhancing or suppressing it, depending on the composition of the exosomal cargo and the cellular context. The impact of exosomes on inflammasome activation is highly complex, with studies indicating both pro-inflammatory and anti-inflammatory effects based on their cellular origin and the specific TME conditions. Exosomes derived from diverse cell types—including immune, epithelial, cartilage, and cancer cells—can modulate inflammasome activity, with immune cell-derived exosomes being the most extensively studied [25]. This review will provide comprehensive insights into the TME, focusing on the dynamic interplay between the immune system and tumors. Moreover, the importance of TILs and the role of exosomes as promising and safer strategies for cancer therapy will be explored. Moreover, this review will also highlight the role of the inflammasome and its activity modulation as a potential therapeutic strategy for cancer treatment [26].

## 2. Differences in Cell Behavior Between Normal and Cancerous Conditions

Table 1 summarizes the differences in the behavior of different immune cells under normal and tumor conditions and their impact on tumor progression.

## 3. Cancer Immunoediting

The interaction between the immune system and cancer cells is distinguished by three different phases (Figure 1). In the first phase, referred to as the elimination phase, cancer cells are actively targeted and removed through a process known as immunosurveillance. This phase involves the collaborative efforts of innate and adaptive immunity, working together to recognize and eliminate altered cells before they can manifest clinically. The equilibrium follows, during which transformed cells persist but are held in check and under the control of the immune system. During this stage, the immune system acts as a regulator, with adaptive immunity playing a crucial role in limiting the proliferation of clinically undetectable occult tumor cells while also modifying the immunogenic properties of tumor cells. The final phase (the escape phase) signifies the point at which malignant cells successfully evade immune control, leading to tumor progression. The regulation of these processes heavily relies on the involvement of leukocytes and cytokines. Within a tumor, a wide range of immune cell types can be found, involving neutrophil granulocytes, macrophages, mast cells, DC, NK cells, naive and memory lymphocytes, B cells, and effector T cells, including Th 1, 2, and 17 cells, Tregs, T follicular helper cells and cytotoxic T cells [47].

## 4. Phenotype of Tumor-Infiltrating Lymphocyte

Figure 2 displays the different phenotypes of TILs, including T-cells and B-lymphocytes. The clinical outcome is influenced by both the extent of lymphocytic infiltration and the specific infiltrate phenotype. A favorable prognosis is associated with type 1 T-cells, where CD4+ Th1 cells play an essential role in antigen presentation through cytokine secretion. Additionally, CD8+ cytotoxic T-helper (CTL) cells are crucial for tumor elimination [48]. In contrast, type 2, CD4+ Th2 can inhibit CTL activity, promote the proliferation of B-lymphocytes, and may activate an anti-inflammatory immune response that could potentially favor the progression of tumor [49,50].

## 5. Tumor-Infiltrating Lymphocyte (TIL) Therapy

To date, there are three types of adoptive cell therapy (ACT): chimeric antigen receptor T-cell (CAR-T) therapy, TIL therapy, and T cell receptor (TRC) therapy [52]. The last one is the least used due to being restricted by the patient’s major histocompatibility complex (MHC) type, high autoimmunity risk, and complexity of engineering [53]. Both CAR-T and TIL are used to treat patients in advanced stages of cancer, including those who are resistant to checkpoint inhibitor therapies. CAR-T therapy has been approved for hematological cancer like acute lymphoblastic leukemia, chronic lymphocytic leukemia, multiple myeloma, and different forms of lymphomas. TIL has shown promising results in treating certain solid tumors [54,55,56,57,58]. This review will focus on this last type of ACT for being directly connected with the TME. 

TILs are immune cells, mostly T cells, that migrated from the bloodstream into the TME to recognize and eliminate tumor cells. In ACT, TILs are harvested from the patient’s tumor, expanded ex vivo in the laboratory, and reinfused back into the patient. The equilibrium between pro-tumor and anti-tumor responses of TIL is predominantly influenced by the TME, which varies from patient to patient. Factors like immunosuppressive cytokines like TGF-β, immune checkpoint molecules as PD-1 and CTLA-4, TAMs, Tregs, or hypoxia in TME can impair the efficacy of TIL [59]. A key step in TIL therapy that eliminates some of these factors and increases the expansion and functionality of the infused TILs is lymphodepletion. Several findings suggest that lymphodepletion can enhance TIL effectiveness through multiple mechanisms that include the removal of Tregs elevation in host homeostatic cytokines such as IL-7 and IL-15 and reduction in endogenous lymphocytes. This reduction reduces competition for these critical trophic cytokines among the host’s lymphocytes [60,61]. Additionally, the adoptively transplanted T lymphocytes are regulated by APCs that activate because of lymphodepletion.

A non-myeloablative (NMA) lymphodepletion regimen is normally used before TIL injection. This regimen typically involves lower doses of chemotherapy or low-dose total body irradiation (TBI). Ongoing research is actively refining lymphodepletion protocols to avoid toxicity and improve patient outcomes in different types of cancer [62,63,64,65,66]. An NMA lymphodepletion regimen with cyclophosphamide and fludarabine caused a 50% objective response rate in patients with metastatic melanoma treated with TIL. A strong long-term persistence of the adoptively transferred cells was also reported [67,68]. Although melanoma has been the main focus of research in this field, there is clinical evidence suggesting that TIL therapy with lymphodepletion may be beneficial for other types of cancer, among them non-small cell lung cancer (NSCLC), ovarian cancer, cervical cancer, colorectal cancer, and renal cell carcinoma (RCC). TIL therapy has also been combined in the clinic with immune checkpoint and BRAF inhibitors [69,70,71,72,73,74]. 

## 6. Combination Therapy with TIL

### 6.1. Immune Checkpoint Inhibitors

Recent trials have shown promising preliminary results with the combination of TIL treatment and anti-PD-1/PD-L1 antibody therapy [75]. T lymphocytes express immunological checkpoint receptors on their surface, including CTLA-4 (CD152) and PD-1/PD-L1. In cancer patients, CTLA-4 and PD-1 molecules on effector T cells are upregulated, binding to B7-1/B7-2 and PD-L1 on APCs or tumor cells. This binding leads to the inhibition of T cell function, a blockade that can be relieved through the use of anti-CTLA-4 and anti-PD-1 antibodies [76]. The first small molecule inhibitor based on the PD-1/PD-L1 axis, a derivative of the antibiotics sulfamethoxine and sulfamethimazole, was reported [77]. In advanced triple-negative breast cancer (TNBC) patients, the anti-PD-L1 drug atezolizumab has significantly reduced PD-L1-positive (PD-L1+) metastases and increased overall survival [78]. PD-1 inhibitors include pembrolizumab (Keytruda), nivolumab (Opdivo), and cemiplimab (Libtayo), while PD-L1 inhibitors include atezolizumab (Tecentriq), nivolumab (Bavencio), and durvalumab (Imfinzi). Both PD-1 and PD-L1 inhibitors are effective in treating various cancer types. Additionally, CTLA-4 inhibitors like ipilimumab (Yervoy) and tremelimumab (Imjuno) are monoclonal antibodies that bind to CTLA-4, inhibiting its activity and leading to the activation of TILs [79,80].

### 6.2. BRAF Inhibitor

The BRAF gene plays a crucial role in cell development and differentiation. In various malignancies, BRAF mutations disrupt the ERK/MAPK signaling pathway, leading to increased cell proliferation. Activating BRAF mutations, primarily V600E, can trigger immune-escape mechanisms, hindering cells less responsive to T-cell immune responses. Notably, the BRAF inhibitor vemurafenib can diminish linked immunosuppressive signals, enhance lymphocyte infiltration, and reduce the prevalence of immunosuppressive cells [81,82].

## 7. History of Exosomes

Exosomes are microscopic extracellular vesicles (EVs) that emerge from early endosomes (Figure 3). Exosomes were first found in the maturing mammalian reticulocyte by Stahl and their team in 1983 and subsequently by Johnstone and colleagues in the same year. Hence, initial reports on exosomes emerged in the mid-1980s [83,84]. In the beginning, researchers believed exosomes were only “garbage bags” used for disposing of undesired components [85]. However, accumulating evidence suggests that exosomes play a significant role in various cellular processes, providing a unique mode of intercellular interaction and influencing both pathological and physiological functions [86,87]. In 1991, Rose Johnstone demonstrated the presence of both the nucleoside transporter and the transferrin receptor in exosomes [88]. The researchers noticed that specific cellular stressors can lead to the internalization and shedding of these membrane components at different periods [89]. Several publications from the 1980s and 1990s focused on quantifying EVs and highlighted the changes in EV levels in various diseases. For example, Lee et al. [90] study on elevated microparticles in temporary brain ischemia and other infarctions marked the phenomena, which have since been examined in conditions such as angina [91] and Crohn’s disease [92]. Researchers also identified the capacity of immune cell EVs to display antigens [93], leading to the use of EVs as anti-tumor vaccines. Indeed, this work motivated the Amigorena lab to investigate whether DCs release EVs that, once loaded with tumor peptides, can effectively target tumors [94] paving the way for clinical trials in the subsequent decade [95]. This was a significant development as it highlighted the potential of EVs to participate in biological processes. Together, the recognition that EVs might have physiological roles, play as biomarkers, and hold therapeutic potential sparked a surge of interest in EVs in the early twenty-first century.

Early reviews on EV biology emerged in the decade following the 2000 [96]. Researchers delved deeper into EV nature, examining their proteome and lipidome across various cell types [97]. Mackenzie and colleagues [71] highlighted the crucial role of immune cell-produced EVs in immune system function [98]. The growing interest in EVs allowed for their exploration of anti-tumor therapies [99] and a better understanding of their immune system functions [100]. Remarkably, functional nucleic acid transfer was demonstrated [101], and EV-mediated communication was observed in plant cells as well [102]. With the rising interest in EV-based and stem cell therapies in 2009 [103], studies on vesicles derived from mesenchymal stem cells multiplied, broadening EVs’ therapeutic possibilities. In contrast, the signals received by the cell of origin can impact exosome synthesis and content. One assumption proposes that tumor cells can adapt to a hypoxic microenvironment by releasing exosomes that stimulate angiogenesis or facilitate metastasis to more tumor-favorable environments. This assumption received subsequent support from various pieces of evidence [104].

**Figure 3 ijms-26-02716-f003:**
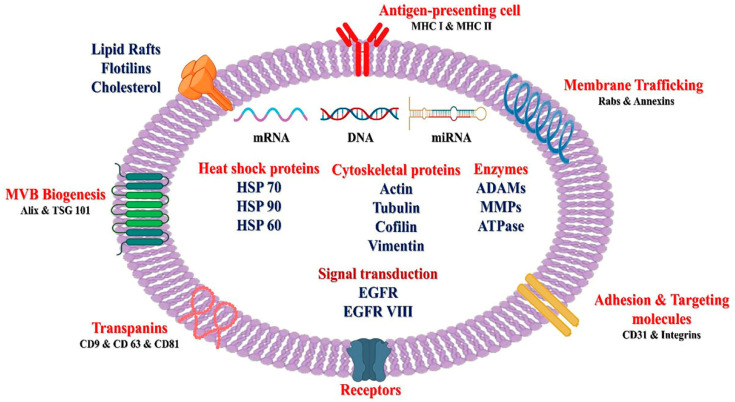
Typical structure of exosomes, nanosized extracellular vesicles enclosed by a phospholipid bilayer, encompassing various proteins (Integrins, TSG 101, Alix, HSP), nucleic acids (DNA, mRNA, miRNA), and other receptors. The figure is adapted from Kumar et al. [105].

## 8. Exosomal Role in Cancer Therapy

Exosomes, nanometer-sized vesicles enclosed by a lipid bilayer (Figure 3), are released by various cell types and are present in most bodily fluids, including breast milk, blood, saliva, bile, urine, pancreatic juice, and peritoneal and cerebrospinal fluids [105,106]. Exosomes exhibit a tenfold greater propensity for internalization and adhesion to tumor cells in comparison to liposomes of corresponding size, which reinforces their enhanced suitability for cancer-targeting [107]. In addition, exosomes tend to accumulate more in tumor tissues with inadequately developed blood vessels than in healthy tissues due to elevated permeability and retention effects. This characteristic enables exosomes to distribute drugs more efficiently, effectively reaching most solid tumors. Moreover, exosomes can be modified to transport tumor-targeting peptides, antibodies, or proteins for the precise delivery of drugs and therapeutic nucleic acids. These qualities firmly establish exosomes as formidable candidates for targeted cancer therapy.

Recently, scientists improved the precision of targeting tumors by employing a magnetic method. They achieved this by connecting superparamagnetic-conjugated transferrin to blood exosomes with transferrin receptors along with applying an external magnet at the tumor site. In particular, Qi and his research team created magnetically guided exosomes, resulting in suppressing the tumor growth effectively [108].

In contrast to the administration of free drugs in animal tumor models, the delivery of chemotherapeutics-loaded exosomes has shown significant enhancements in their anti-tumor effects. For example, drugs such as Doxorubicin, Paclitaxel, and Withaferin, which are frequently utilized in cancer treatment, can lead to adverse effects on animal tissues. Nevertheless, when these drugs are encapsulated in exosomes (exosome-delivered chemotherapeutics), they mitigate the side effects and deliver a more potent and prolonged therapeutic impact [15,109,110].

## 9. Dualist Actions of Exosomes in Carcinogenesis

Exosomes serve as a natural transporter of a wide array of bioactive compounds derived from donor cells, involving lipids, proteins, and different types of RNAs such as long non-coding RNAs (lncRNA), microRNAs, inhibitors, and antibodies [106]. Exosomes can transmit these molecules to nearby as well as distant recipient cells. Consequently, exosomes garnered significant attention as a potential natural carrier of cancer therapies. The impact of exosomes on cancer growth and progression varies depending on their cellular source, influencing the course of the disease. Exosomes secreted by cancer cells are implicated in shaping the pre-metastatic microenvironment, facilitating tumor advancement, immune evasion, anti-apoptotic signaling, angiogenesis, treatment resistance, and various other processes [111,112,113]. In contrast, exosomes originating from normal cells, such as T and B lymphocytes and DCs, play a critical role in preventing tumor progression [114,115].

## 10. The Origin of Exosomes

The source of exosomes dictates whether they promote or inhibit tumor growth, as illustrated in Figure 4.

### 10.1. Exosomes Released from Tumor Cells

Exosomes originating from tumors can exhibit diverse functional roles, contingent upon their cellular origin and the surrounding environmental conditions. For instance, exosomes derived from rat pancreatic adenocarcinoma have been observed to activate cytotoxic T cell (CTL) responses specific to tumor antigens while simultaneously inhibiting leukocyte proliferation by downregulating the Zeta chain of T cell receptor-associated protein kinase 70 (ZAP70) and extracellular signal-regulated kinases 1,2 (ERK1,2) [117]. Additionally, exosome proteins depleted of miRNA can function as agonists, preferentially activating DCs and cytokine-induced killer cells [118]. The presence of leukemia cell-derived exosomes induces the secretion of TNF and IL-12p70 by DCs [119], promoting T cell proliferation and elevating IFN levels, thereby enhancing cytotoxic T lymphocyte (CTL) activity via the FasL/Fas signaling pathway in renal cancer [120]. On the contrary, exosomes derived from breast cancer can impair NK cell cytotoxicity and hinder the proliferation of CD8+ and CD4+ T cells, potentially diminishing the immune system’s effectiveness in fighting cancer [121]. Moreover, tumor-released exosomes in head and neck cancer have been demonstrated to induce a suppressor phenotype in CD8+ T cells, primarily through the cooperative actions of exosomal components like RNA and galectin1 [122]. Additionally, exosomes derived from melanoma cells alter the transcriptome of CTLs, ultimately diminishing cytotoxic immune responses [123].

### 10.2. Exosomes Derived from DCs

The DCs hold a pivotal role in cancer immunotherapy due to their capability to capture and display tumor-associated antigens, making them essential contributors to tumor immunity. However, their effectiveness in combating low-immunogenic tumors has remained unsatisfactory. Implications such as the activation of Tregs, limited antigen uptake efficiency, and antigen availability have posed challenges [124]. Recent research suggests that exosomes can serve as an ideal source of antigens for DC vaccinations, emphasizing the need to explore how exosome-based DC vaccines generate anti-tumor immunity before establishing their suitability as tumor antigens for DC vaccine-based immunotherapy [125].

The DCs are highly efficient APCs known to release exosomes that can trigger potent anti-cancer effects. Exosomes produced by DCs, containing chaperones like MHC I, MHC II, HSP70-90, and CD86, can stimulate CD4+ and CD8+ T cells [126,127]. The secretion of exosomal peptide MHC I is transferred to CD8+ T cells under the influence of released IL-2 and exosomal CD80, leading to increased proliferation of CD8+ T cells and the production of a more robust anti-tumor immune response in vivo [128]. Several studies have observed the activation of CD8+ and CD4+ T cells when exposed to DCs exosomes associated with the induction of an anti-tumor immune response in vivo through exosomal CD80 and endogenous IL-2 [129,130]. Furthermore, in mice with hepatic cell carcinoma, exosomes derived from alpha-fetoprotein-expressing DCs induced the production of more IFN-expressing CD8+ T cells, triggering higher levels of IFN and IL-2, with reduction in CD25+ Foxp3+ Tregs, TGF-β, and IL-10 levels [131].

### 10.3. Exosomes Derived from B Lymphoma Cell

Exosome-based DC vaccines can induce clonal proliferation of T cells through exposure to exosomes derived from diffuse large B cell lymphoma cells [132]. Conversely, exosomes from B cell lymphoma cells have been found to induce apoptosis in CD4+ T lymphocytes via MHC II [133]. B lymphoma cells induced by heat shock released exosomes with elevated levels of HSP90 and HSP60, along with heightened immunogenicity molecules like MHC I, MHC II, CD86, CD40, RANTES, and IL-1. These exosomes effectively stimulate CD8+ T cells, yielding an anti-cancer effect [134]. When DCs interact with exosomes from B cell lymphoma cells, they can enhance the activation of T cells, leading to the release of TNF-α and IL-6, while simultaneously decreasing the production of immunosuppressive cytokines like IL-4 and IL-10 [132].

### 10.4. Exosomes Derived from T Lymphocytes

Immunotherapy is a rapidly emerging and promising treatment approach that utilizes genetically modified T cells to express the chimeric antigen receptor [135,136]. There are two types of T cells, CD8+ cytotoxic T lymphocytes (CTLs) and CD4+ helper T cells. CD8+ CTLs play a crucial role in supporting the body’s defense against intracellular infections and tumor cells by directly binding to antigens through MHC I. In addition to their direct killing of tumor cells, activated CD8+ T cells can indirectly destroy tumor cells by releasing exosomes [137]. In a mouse model of melanoma, intratumoral delivery of exosomes derived from activated CD8+ T cells effectively hindered tumor invasion and metastasis mediated by fibroblastic stroma [138]. It has been discovered that T cell exosomes expressing the CD63 protein contain specific miRNAs that regulate immune responses and immune system development, playing a pivotal role in enhancing interaction between antigen-presenting T cells [139]. CD8+ T cells and CD63+ exosomes produce similar anti-infective effects [140].

Most surface markers of CD4+ helper T cells can interact with MHC II on the surface of APCs to initiate, modify, or support immune responses. Exosomes produced by CD4+ T cells have been established as a fundamental pathogenic mechanism in several inflammatory diseases [141]. These exosomes can employ target cells through CD4–MHC interactions, ultimately leading to the elimination of immune-deficient cells [142]. Exosomes derived from activated CD4+ helper T cells can also act as potent inducers of phagocyte and B cell activation, which support the inflammatory response [143].

### 10.5. Exosomes Derived from NK Cells

NK cells function as the body’s initial defense against various disorders. Exosomes released by NK cells are equipped with cytotoxic proteins such as FasL and perforin, besides characteristic NK markers like CD56 [144]. Additionally, NK exosomes possess the ability to infiltrate tumor tissues directly, enabling them to exert their cytolytic effects. This capability overcomes the challenge of NK cells not naturally homing in on tumor sites [145]. Furthermore, NK cells activate both caspase-independent and caspase-dependent cell death pathways [146].

### 10.6. Exosomes Derived from Myeloid-Derived Suppressor Cell

A diverse group of immature myeloid cells referred to as myeloid-derived suppressor cells (MDSCs) possesses a remarkable ability to inhibit the cytotoxicity of T/NK cells, rendering them a significant obstacle in cancer immunotherapy [147]. Activation of MDSCs under cancerous conditions through utilizing various pharmacological drugs has been extensively investigated. Recent studies have begun to shed light on the immunosuppressive functions of MDSC-derived exosomes within the microenvironment of both cancer and autoimmune disorders [148]. Exosomes derived from MDSCs have been found to carry cargo that matches their role in mediating immunosuppression [149]. In mice bearing breast tumors, augmented MDSC-derived miR-126a+ exosomes have been shown to potentially stimulate metastasis and confer resistance to therapy [150].

### 10.7. Exosomes Derived from Tumor-Associated Macrophage

Macrophages in the TME can dimmish the activity of T cells, paving the way for cancer cells to escape immunity. Remarkably, TAMs often exhibit two competing phenotypes: the anti-tumorigenic M1 subtype and the pro-tumorigenic M2 subtype [151] as previously discussed in this review. Accumulating pieces of evidence suggest that TAMs release exosomes that influence various aspects of cancer biology and the immune response [152]. These exosomes, originated from TAMs, can create an immune-suppressive environment and enhance the progression of ovarian cancer by transferring miRNAs into CD4+ T cells. Moreover, exosomes derived from M2 macrophages transmit oncogenic miRNAs, promoting cancer cell invasion, migration, and resistance to chemotherapy [153]. These TAM-derived exosomes primarily function as markers for the polarization of Th1 and M1 subtypes, containing contents that enhance pro-inflammatory signaling and the immune response [154].

### 10.8. Exosomes Derived from Mast Cells (MCs)

Exosomes, with their essential roles in RNA and protein transfer, intercellular interaction, and immunoregulation, can also be released by MCs. Remarkably, MC-originated exosomes have been demonstrated to compromise intestinal barrier function, likely due to the delivery of miRNAs to targeted cells [155]. Recent research has revealed that lung cancer cells can internalize MC-derived exosomes, subsequently promoting cancer cell growth by transferring the KIT protein [156]. Additionally, the biological processes of DCs, T cells, and B cells can be influenced by exosomes produced by MCs. For instance, exosomes from MCs that express CD63 and OX40L have been found to enhance the communication between OX40L and OX40, resulting in the proliferation and development of CD4+ Th2 cells [157]. Furthermore, MC-derived exosomes stimulate immature DCs to up-regulate molecules such as MHC II, CD40, CD80, and CD86, enabling T cells to present antigens and initiate the development of immune responses targeted against specific antigens. The potential anti-cancer properties of MC-derived exosomes are currently under investigation [98].

### 10.9. Exosomes Derived from Neutrophils

Exosomes released by neutrophils have been demonstrated to adhere to and degrade the extracellular matrix through the actions of neutrophil elastase (NE) and the integrin Mac-1, facilitating the progression of inflammatory diseases [158]. In contrast, Li and colleagues recently made a remarkable discovery, revealing that these exosomes strongly inhibit the proliferation and migration of endothelial cells, thereby hindering pathological angiogenesis in immunological diseases [159]. Additionally, Vargas and co-authors tentatively confirmed the presence of the tumor susceptibility gene 101 in neutrophil-derived exosomes [160]. However, to the best of our understanding, there is a lack of pertinent research to elucidate the underlying molecular mechanisms of neutrophil-derived exosomes in the regulation of anti-cancer immune responses. 

Table 2 summarizes the various sources of exosomes.

## 11. History of Inflammasomes

The term inflammasome was first introduced in 2002 by Dr. Jürg Tschopp [161]. This word describes an intracellular protein complex involved in immune responses. NLRP1 inflammasome was the first complex that was characterized, consisting of the Nod-like receptor (NLR) NLRP1, the adaptor protein ASC (Apoptosis-associated speck-like protein containing a CARD), and inflammatory caspases, for example, caspase-1 and -5. Inflammasomes belong to the Nod-like receptor (NLR) family. This group of intracellular pattern recognition receptors (PRRs) detects cellular stress, pathogens, and damage-associated molecular patterns (DAMPs). The inflammasomes cause inflammatory responses through two mechanisms: one, by activating caspases which process and release pro-inflammatory cytokines, for example, IL-1β and IL-18, and, two, by inducing the process of pyroptosis. The term pyroptosis describes a form of inflammatory cell death characterized both by plasma membrane rupture and cytokine release [162]. Since the discovery of the NLRP1 inflammasome, over twelve different inflammasomes have been characterized, including NLRP1, NLRP2, NLRP3, NLRP6, NLRP7, NLRP9, NLRP12, NAIP/NLRC4, AIM2, IFI16, CARD8, and PYRIN [163]. Most of these belong to the NLR family, with 22 members in humans and 34 in mice. They are characterized by a nucleotide-binding and oligomerization domain (NOD) and a C-terminal leucine-rich repeat (LRR). The family members are classified according to their N-terminal domains, which can include NLRA, BIR, CARD, or Pyrin domains.

NLRP1 and NLRP3 are among the most studied inflammasomes. NLRP3 is activated by a broad range of triggers, among which are MAMPs (microbe-associated molecular patterns) and DAMPs, such as particulate matter, cholesterol crystals, and cellular stress signals [164]. The precise ligand for NLRP3 remains difficult to characterize, although it is known that NLRP3 activation requires post-translational modifications such as ubiquitination, phosphorylation, and interactions with NEK7. On the other hand, the activation mechanisms of other inflammasomes, such as AIM2, NAIP/NLRC4, and Pyrin, are better understood. Cytosolic double-stranded DNA from both microbial and host origins activates AIM2 [165,166]. NAIP/NLRC4 responds to bacterial components such as flagellin and proteins of the bacterial type III secretion system [167,168]. Pyrin senses alterations in cellular dynamics caused by infection-induced modifications to Rho family small GTPases [169].

The study of inflammasomes has increased in recent years and findings have connected inflammasome dysregulation to various diseases, among them cancer. Inflammasomes play a wide range of roles in tumorigenesis and immune responses within the TME. These roles can be both pro-tumorigenic and anti-tumorigenic, depending on the type of cancer, cancer etiology, and cells activating the inflammasome pathway within the TME. This dual role highlights the complexity of inflammasomes in cancer biology, spurring a growing body of research into inflammasome-targeted therapies for cancer treatment [170,171].

## 12. Inflammasomes in Cancer Therapy

Inflammation caused by inflammasomes has been linked to the development and progression of cancer. Inflammasomes trigger inflammatory responses through caspase-1 activation, leading to the release of pro-inflammatory cytokines like IL-1β and IL-18 [17,18,19]. These cytokines influence tumor development by modifying the TME, modulating immune responses, and promoting cancer cell survival and metastasis. Therefore, targeting inflammasomes with specific inhibitors or activators offers a promising therapeutic approach for treating cancer [26]. 

### 12.1. Inhibition of Inflammasomes

The most prevalent therapeutic approach against cancer consists of inhibiting inflammasome activation [172]. The principal target of these attempts to restrict inflammasome activation is NLRP3 [26,173]. Research has shown this inflammasome plays an important role in many cancer types. One of the main challenges researchers face when developing these therapies is determining whether to prioritize the upstream or downstream components of the inflammasomes [26,174,175,176]. The upstream targets, such as NLRP3, could allow for more precise interventions by specifically inhibiting certain inflammasome activations without affecting other sensors like AIM2, NLRP1, and NLRC4, which detect pathogens and other stimuli. On the other hand, downstream targeting of IL-1β and IL-18 could be less selective overall but more effective in suppressing the pro-inflammatory signals that promote cancer progression.

One of the two most studied NLRP3 inhibitors are MCC950 and OLT1177, which selectively block the NACHT domain of NLRP3, avoiding its activation and subsequent inflammatory response. MCC950 has shown promising results in treating various cancer types, including pancreatic cancer, colorectal carcinoma, and head and neck squamous cell carcinoma [177,178,179]. OLT1177 has been used against melanoma [180,181]. Similarly, other inhibitors like CY-09, tranilast, and oridonin also target the NACHT domain to inhibit inflammasome activation. Some of these treatments also show anti-tumor effects in preclinical models [178,182,183,184,185]. Although NLRP3 inhibitors have worked to reduce inflammation, their combination with other therapeutic methods, such as immune checkpoint inhibitors, such as anti-PD-1, may enhance their anti-tumor effects by modulating the TME [186,187]. For example, OLT117 has been shown to disrupt the IL-1β/IL-6/STAT3 axis in the TME, which decreases the immunosuppressive activity of MDSCs and improves anti-tumor immunity when combined with anti-PD-1 therapy [181].

Besides NLRP3 inhibitors, other strategies prioritize the targeting of other inflammasome components or proteins involved in their activation. For example, glycyrrhizin has been shown to inhibit both NLRP3 and AIM2 inflammasomes [188]. Methylene blue is a broad-spectrum inflammasome inhibitor that affects multiple inflammasome pathways, among them NLRP3, NLRC4, and AIM2 [189]. Additionally, andrograholide has shown promising results in inhibiting AIM2 in the prevention of colitis-associated cancer [190]. AIM2 and NLRC4 are involved in pathogen detection as well as immune cell activation. Targeting the NLRP3 inflammasome, therefore, could offer a more selective therapeutic approach, which reduces the risk of unwanted suppression of the immune system [191].

Alternatively, another approach to targeting inflammasomes consists of inhibiting caspase-1, the downstream effector that mediates pyroptosis. The caspase-1 inhibitor VX-765 has shown anti-tumor effects through the inhibition of pyroptosis and reduction in IL-1β secretion in non-small cell lung cancer (NSCLC) models [192]. The effects of caspase-1 inhibition on tumor growth, however, are context-dependent. Some studies seem to indicate that inhibiting pyroptosis could allow cancer cells to evade immune surveillance. Others show potential benefits in specific immune subsets such MDSCs [193,194,195,196].

### 12.2. Activation of Inflammasomes for Cancer Immunotherapy

Although most inflammasome-targeting strategies have focused on inhibition, an alternative approach to cancer therapy may be the opposite method: the activation of inflammasomes. This approach may work because inflammasome activation can trigger pyroptosis in cancer cells, releasing pro-inflammatory cytokines such as IL-1β and IL-18 that can aid in anti-tumor immune responses. The compounds polyphyllin VI and 17β-estradiol have been shown to activate NLRP3 in cancer cells, inducing pyroptosis and improving the anti-tumoral immune response in NSCLC and hepatocellular carcinoma [192,197,198]. Furthermore, recombinant IL-18 has been used in clinical trials as an immunotherapy to enhance the activation of NK cells and T cells, which are critical for the recognition and elimination of tumor cells [191]. The development of specific inflammasome activators, however, remains an area of active research.

A promising strategy consists of using a combination of inflammasome activators with immune checkpoint inhibitors. For example, the addition of inflammasome activation may improve the efficacy of treatments like anti-PD-1 therapies by increasing the anti-tumor immune response. This combination approach attempts to boost the immune system’s ability to recognize and destroy cancer cells simultaneously while also triggering pyroptosis in tumor cells [199].

## 13. Dual Actions of Inflammasomes in TME

Inflammasomes can be activated in the diverse subgroups of cells in TME, including tumor cells, TAMs, tumor-associated fibroblasts, and marrow-derived suppressive cells [20,21,22,23]. Recent findings have revealed the dual role of inflammasomes in TME, promoting or inhibiting tumor progression depending on different inflammasomes and tumors. Inflammasomes are mainly involved in tumorigenesis, metastasis, and immune evasion of malignant tumors [24].

Some inflammasome mutations have shown a tumorigenesis effect. Gain-of-function mutations in NLRP1 are associated with multiple self-healing palmoplantar carcinoma [200]. Individuals with NLRP1 variant rs12150220 or NLRP3 variant rs35829419 are more susceptible to nodular melanoma [201]. The NLRP3 variants rs10754558 and rs4612666 are linked with gastric cancer, and the amino acid mutation Q705K is related to pancreatic cancer [202,203]. It remains to be determined whether the pro-tumorigenic effects of these inflammasome mutations are driven by chronic inflammation. Transgenic mice carrying the relevant mutations could provide valuable insights into the underlying mechanisms. Suppression of inflammasomes has proved to attenuate tumorigenesis. For instance, in glycoprotein 130 (gp130)^F/F^ mice with spontaneous intestinal-type gastric cancer, the absence of ASC prevents tumor formation. This ASC knockout leads to decreased levels of mature IL-18 in the gastric tumor epithelium, which in turn enhances caspase-8-mediated apoptosis [204]. Likewise, knocking out ASC, inhibiting caspase-1, or deleting germ cells all reduce the occurrence of spontaneous cecal carcinogenesis in AhR-knockout (AhR^−/−^) mice. These studies indicate that inflammation triggered by bacteria and inflammasome activation play harmful roles in tumor development [194]. Supporting this, mice with overexpressed IL-1β in the stomach develop spontaneous gastric inflammation and cancer [205].

In contrast, other research has proposed that inflammasomes may play a protective role in tumor development. For example, the reduction in NLRP3 inflammasome components has been observed during the progression of multistage hepatocarcinogenesis [206]. ASC and caspase-1 recruited immune cells during tumorigenesis of chemically induced squamous cell carcinoma [195]. Mice without ASC, caspase-1, or NLRP3 had more severe colitis and tumorigenesis in colitis-associated cancer models [207,208]. The attenuated hematopoietic cell-derived IL-1β and IL-18 at the tumor site of Nlrp3-knockout (Nlrp3^−/−^) mice are found to be the key to inflammation and tumorigenesis. These studies align with results from pyrin knockout mice, which also exhibited more serious colitis and an increased tumor burden [209]. Mice lacking caspase-1 or NLRC4 exhibited greater proliferation of colonic epithelial cells and decreased apoptosis of tumor cells, which led to an increase in tumor formation in colitis-associated colorectal cancer models [193].

The role of inflammasomes in tumorigenesis may vary as malignant tumors progress. While NLRP3 inflammasome components are upregulated in tissues affected by hepatitis and cirrhosis, their expression is reduced in hepatocellular carcinoma [206]. ASC knockdown has opposite effects on tumor development in melanoma, suppressing tumorigenesis in metastatic melanoma and promoting it in primary melanoma. This contrasting behavior can be attributed to divergent NF-κB activity downstream of ASC, where it is inhibited in primary melanoma but enhanced in metastatic melanoma [210]. The effect of inflammasomes in tumorigenesis not only depends on the clinical stage but also the cell class in TME. Deleting ASC conditionally in myeloid cells reduces the incidence of chemical-induced skin cancer, whereas eliminating ASC specifically in keratinocytes leads to an increase in tumor formation [211].

Tumor growth can be influenced by GSDMD, alongside IL-1 family members. Elevated GSDMD levels are associated with advanced TNM stages in NSCLC patients, and its knockdown inhibits tumor growth by activating the mitochondrial apoptotic pathway and suppressing the EGFR/AKT signaling pathway [212]. In contrast, GSDMD expression is lower in gastric cancer cells and tissues, where reduced levels of GSDMD promote tumor cell proliferation by accelerating the S/G2 phase transition of the cell cycle. Moreover, GSDMD expression negatively correlates with the activation of STAT3, ERK, and PI3K/AKT pathways [213]. The differences in how GSDMD affects tumor pathways in various cancers may clarify these conflicting results.

Myeloid cells have been the main focus of research to study the connection between inflammasome activation and metastasis since IL-1β is mostly produced by myeloid cells [214,215,216]. This connection has also been found in cancer-associated fibroblasts and tumor cells [22,217]. Even though IL-1β is considered a marker of M-1-like macrophages that activates a tumor-targeted immune response, anomalous inflammasome activation in TAMs causes metastasis in different tumors. A direct relationship between the activation of inflammasomes, in particular NLRP3, and the late clinical stages of metastasis has been proved through clinical data. These data indicate a poor survival rate in patients with breast cancer and lung cancer [21,218]. The use of anakinra or canakinumab blocked the IL-1 signal and inhibited the metastasis of breast cancer [219].

Inflammasome activation can also inhibit metastasis [220]. Interestingly, NLRP3-produced IL-1β induces the migration of colorectal cancer cells, and its activation in liver macrophages (Kupfer cells) decreases the metastasis of colorectal cancer cells. The secretion of IL-18 by NLRP3 in Kupffer cells plays a key role in promoting NK cell maturation and enhancing their tumor-killing activity [221,222]. These results indicate that the NLRP3 inflammasome signaling may either promote or inhibit tumor metastasis, depending on the tumor type and tissue involved. Variations in IL-1β and IL-18 production across different cell subsets could potentially explain the differences observed in downstream events [191].

Immune evasion is quintessential for tumorigenesis and metastasis. Tumor cells can disrupt the Fas receptor, increase the expression of PD-L1, and decrease the expression of MHC-I. In the TME, immunosuppressive factors such as M2-like macrophages, MDSCs, and Tregs are key players that promote immune evasion by tumor cells [223,224]. Various stimulators can trigger the expression and activation of inflammasome components in cancer cells, fibroblasts, and macrophages [22,23,225,226,227,228]. This activation leads to the release of IL-1β and IL-18, which, in turn, influences the expression of PD-L1 on tumor cells and contributes to the recruitment of immune-suppressive cells within the TME [191].

The activation of NLRP3 is associated with immunosuppressive cells such as Tregs, MDSCs, and TAMs. This process can be inhibited with NLRP3 inhibitors. The production of IL-18 in the TME of myeloma multiple has been linked to an increase in MDSCs, decreased T cell activity, and worse survival [229]. In other scenarios, IL-1β and IL-18 are known to enhance T cell-mediated anti-cancer immunity [230,231]. Specifically, IL-18 derived from CD4+ T cells or introduced exogenously has been shown to promote the proliferation and anti-tumor activity of CD8+ T cells and CAR-T cells. These conflicting observations may be clarified by measuring the actual concentrations of IL-1β and IL-18 in the TME rather than simply noting their relative changes. The hypothesis is that IL-1β and IL-18 may induce different immune responses at varying concentrations. Moreover, understanding the specific locations of inflammasome, IL-1β, and IL-18 expression could be crucial. Investigating inflammasome activation across different cell subsets might provide deeper insight into the inflammatory network and immune regulation within the TME, paving the way for targeted therapeutic strategies [191].

Both tumor-derived and fibroblast-derived inflammasome activation allows for an immune-suppressive environment in most cases, while myeloid cell-derived inflammasome activation seems to be favorable for anti-tumor immunity [227,230,232]. One potential explanation could lie in the varying size and duration of inflammasome activation between myeloid cells and other cell types within the TME [191]. Studies have shown differences in inflammasome activation between macrophages and neutrophils, suggesting that inflammasomes can be activated to different extents, leading to distinct downstream effects. Differences in the signals exchanged between myeloid cells and other cell types could also play a role. For example, in DCs and macrophages, the release of IL-1β and IL-18 facilitates antigen presentation and T-cell recruitment. Conversely, in tumor cells, inflammasome activation is frequently associated with the secretion of immunosuppressive signals like PD-1/PD-L1 [199,230,233]. Further research is needed to explore how inflammasome activation and its effects differ among cell subsets and to better understand the interactions between IL-1 family signals and other signaling pathways.

## 14. Connection Between Exosomes and Inflammasome Roles in TME

The interplay between exosomes and inflammasomes in the TME is a subject of growing interest due to its implications in cancer progression, inflammation, and immune response regulation. Exosomes have been identified as key mediators in the communication between tumor cells, immune cells, and other components of the TME. This communication can enhance or suppress inflammasome activation depending on the nature of the exosome cargo and the cellular context. The role of exosomes in inflammasome activation is complex, with evidence supporting both pro-inflammatory and anti-inflammatory effects depending on the origin of the exosomes and the specific context of the TME. Exosomes derived from various cell types, including immune, epithelial, cartilage, and cancer cells, have been shown to modulate inflammasome activity, with immune cell-derived exosomes being the most extensively studied [25]. 

### 14.1. Exosome-Mediated Inflammasome Activation in Immune Cells

Exosomes derived from immune cells, particularly macrophages, have been shown to significantly influence inflammasome activation within the TME [234]. These exosomes carry a variety of bioactive molecules, such as cytokines, RNA, and signaling proteins, that can activate inflammasomes in recipient cells. Upon uptake by recipient cells, these immune cell-derived exosomes trigger NF-κB activation, leading to the upregulation of inflammasome components like NLRP3, pro-IL-1β, and pro-IL-18. The dissociation of NF-κB from IκB in these cells promotes its nuclear translocation, where it initiates the transcription of these pro-inflammatory molecules, thus enhancing the inflammatory response in the TME [235]. Furthermore, exosomes from immune cells promote inflammasome complex formation through the interaction of PRRs, ASC, and pro-caspase-1, leading to the activation of caspase-1 and the subsequent secretion of pro-inflammatory cytokines such as IL-1β, IL-18, and TNF-α.

In the context of cancer, immune cell-derived exosomes are particularly potent in inducing inflammasome activation. Studies have demonstrated that exosomes from macrophages in glioblastoma multiforme (GBM) patients contain miRNA-21, which inhibits the expression of the tumor suppressor gene PDCD4 in recipient GBM cells. This inhibition contributes to tumor cell proliferation and promotes chemoresistance, especially to treatments such as temozolomide [236]. Moreover, exosomes from immune cells also facilitate tumor immune evasion by modulating inflammasome activation, which can shift the immune response toward a more suppressive state. The capacity of immune cell-derived exosomes to modulate both inflammation and immune tolerance is a critical factor in shaping the TME and influencing cancer progression [237,238,239].

### 14.2. Cancer Cell-Derived Exosomes and Inflammasome Modulation

Exosomes derived from cancer cells have also been implicated in inflammasome activation. For instance, exosomes released from the HepG2 cell line, a hepatocellular carcinoma model, in response to palmitate fatty acid treatment, promote pro-IL-1β expression and the release of mature IL-1β from THP-1 monocytic cells [240]. Similar findings have been reported with exosomes derived from malignant ascites and amniotic fluid [241]. These findings suggest that exosomes from cancer cells not only enhance inflammasome activation but also propagate a pro-inflammatory environment, contributing to the inflammatory milieu that promotes tumor growth and metastasis.

Cancer-derived exosomes can also orchestrate the polarization of tumor-associated macrophage TAMs, further contributing to the inflammatory response in the TME [242,243,244]. M1-like TAMs, induced by pro-inflammatory signals, are associated with tumor growth and angiogenesis. M2-like TAMs are typically involved in immune suppression and tissue remodeling. Cancer-derived exosomes can shift TAMs toward the M1-like phenotype, enhancing the pro-inflammatory state of the TME and supporting tumor progression [245]. This polarization is linked to poor clinical prognosis [246,247,248]. A recent study showed that exosomes derived from glioma cells, enriched with HMGB3, promote M2 polarization in macrophages and activate the NLRP3 inflammasome, inducing pyroptosis. This mechanism plays a critical role in establishing an immunosuppressive and inflammatory TME in glioma [249]. The NLRP3 inflammasome is pivotal in TAM polarization since its activation is essential for IL-1β production in macrophages and other immune cells. Cancer cells can influence key molecules in the NLRP3 pathway, resulting in TAM reprogramming toward a pro-inflammatory state that supports tumor progression. For instance, exosomes derived from murine lung cancer cells express the tripartite motif-containing protein TRIM59, which promotes the ubiquitination and degradation of ABHD5. This, in turn, activates the NLRP3 inflammasome pathway, leading to IL-1β release from macrophages and supporting the growth and metastasis of lung cancer [250]. ABHD5 plays a crucial role in lipid metabolism, and its absence has been found to enhance macrophage reprogramming and inflammasome activation, further highlighting the connection between metabolic alterations in the TME and inflammasome activation [251]. Additionally, less studied inflammasomes like NLRP6 have been identified as critical for the M2 polarization of macrophages induced by small cell lung cancer (SCLC)-derived exosomes. This process facilitates SCLC metastasis both in vitro and in vivo [252]. These findings provide new insights into the role of novel inflammasome proteins in cancer progression and suggest their potential as therapeutic targets for managing cancer metastasis.

Beyond their role in TAM polarization, cancer-derived exosomes can also influence inflammasome activation in response to external factors such as viral infections. Exosomes secreted by virus-infected cells can carry viral proteins that impact inflammasome activation in neighboring cells. For example, Epstein–Barr virus (EBV)-infected B-cells and nasopharyngeal carcinoma (NPC) cells secrete exosomes containing latent membrane protein 1 (LMP1), a viral protein that enhances B-cell proliferation, tumor growth, and radioresistance. Upon uptake by non-infected cells, LMP1-containing exosomes contribute to the inflammatory microenvironment and promote tumor progression by facilitating immune evasion mechanisms that are characteristic of cancer [253,254]. These findings highlight the role of viral infections in modulating exosome–inflammasome interactions and influencing cancer-related inflammation.

Additionally, cancer therapies can influence exosome–inflammasome interactions. For instance, exosomes released from HepG2 cells treated with ezetimibe do not induce inflammasome activation or IL-1β secretion, suggesting that exosomal cargo composition is sensitive to the metabolic state of the tumor cells [240]. On the other hand, doxorubicin, a commonly used chemotherapeutic agent, has been shown to induce an inflammatory response in cardiomyocytes by activating TLR4, NLRP3, caspase-1, and IL-1β. Exosomes derived from embryonic stem cells (ESCs) were found to mitigate the inflammasome-activating effects of doxorubicin, while exosomes from mouse embryonic fibroblasts (MEFs) had no such effect. This observation suggests that the type of exosomes and their cargo can influence inflammasome activation, indicating a potential therapeutic avenue for modulating the inflammatory responses induced by chemotherapy [255]. The differential effects of exosomes from various cell types underscore the cell-specific nature of exosome cargo and its impact on inflammasome signaling.

### 14.3. Exosome Cargo and Its Role in Inflammasome Activation

The cargo contained within exosomes is a critical determinant of their impact on inflammasome activation and their role in cancer biology. Exosomes derived from different cell types carry distinct biomolecules, including proteins, miRNAs, and lipids, which can influence the inflammatory response in recipient cells [256,257,258]. For example, exosomes from colorectal cancer cells, particularly those harboring KRAS mutations, exhibit distinct miRNA profiles linked to the KRAS gene’s mutation status. These mutations in KRAS have been shown to influence the sorting of miRNAs into exosomes, which then modulate gene expression in recipient cells, contributing to inflammation and tumor progression [259].

Specific proteins involved in exosome biogenesis and cargo sorting, such as CD63 and RILP, play a significant role in determining the molecular content of exosomes [256,259,260]. In the case of EBV-infected cells, CD63 has been shown to mediate the loading of LMP1 into exosomes, which is essential for the viral protein’s pro-inflammatory and tumorigenic effects [261]. Similarly, RILP regulates the packaging of miR-155 into exosomes derived from hepatitis C virus (HCV)-infected hepatoma cells, suggesting that cellular factors can selectively control the contents of exosomes, influencing their ability to modulate inflammasome activation and inflammatory pathways [262].

The selective packing of miRNAs and proteins into exosomes is not fully understood, but several studies have provided insights into the mechanisms behind this process. Exosome secretion and cargo sorting are influenced by specific cellular conditions such as hypoxia, extracellular acidification, and changes in intracellular calcium levels [263,264]. These conditions are commonly found in the TME and can stimulate exosome release, thereby facilitating the transfer of inflammatory signals between cells. Exosome release can also be triggered by the activation of immune pathways, such as those mediated by LPS-stimulated TLR4, which further underscores the link between immune responses and exosome-mediated inflammasome activation in the TME [265].

### 14.4. Exosomal Secretion and TME

The TME, characterized by its inflammatory and immune landscape, is crucial in modulating exosome secretion. Several factors inherent to the TME, such as hypoxia, acidity, and elevated calcium levels, have been shown to enhance exosome release. In particular, the acidic environment of the TME, which results from increased metabolic activity in tumor cells, has been demonstrated to promote exosome secretion from metastatic melanoma cells [266]. This observation suggests that the TME is not a passive environment but actively participates in shaping the nature of exosome-mediated communication, particularly concerning inflammasome activation and the inflammatory response.

The role of exosome secretion in cancer inflammation is further emphasized by the activation of inflammasomes in response to tumor-associated signals. For example, elevated extracellular ATP levels in the TME can activate the P2X7 receptor, leading to an increase in cytoplasmic calcium levels that subsequently promote exosome release. This pathway provides an additional link between inflammation, immune activation, and exosome secretion, highlighting the complex interplay between these processes in the TME [267]. Moreover, inflammasome activation, whether through exosomal delivery of inflammasome components or as a consequence of TLR4 signaling, may further enhance exosome secretion, creating a feedback loop that perpetuates inflammation within the tumor.

## 15. Conclusions, Challenges, and Future Directions

Exosomes have emerged as promising natural carriers for cancer therapies. Their impact on cancer growth and progression varies depending on their cellular origin, shaping the trajectory of the disease by influencing key processes within the TME. Exosomes can be used to enhance TIL function by reversing T-cell exhaustion or delivering immune-stimulating molecules. Furthermore, exosome-based strategies may help improve TIL persistence and infiltration into tumors, making TIL therapy more effective. While TILs and exosome therapy offer exciting opportunities to modulate the TME for improved cancer immunotherapy, several challenges remain. Addressing these hurdles will be critical for translating these strategies into effective clinical applications.

Exosomes reflect their parent cells and the conditions under which they were formed, but identifying their origins in vivo is challenging. They carry diverse molecular cargo, including proteins, RNAs, and lipids, but pinpointing the specific effects of miRNA clusters and individual miRNAs is complex. The mechanisms behind exosome targeting, uptake, gene expression alterations, and physiological effects remain unclear. Moreover, determining their specific effects and dominant mechanisms is difficult due to the presence of multiple molecular components that may contribute to overlapping or distinct functions [268].

Furthermore, exosomes are quickly cleared by macrophages, reducing their circulation time and therapeutic efficacy, while extending their half-life may cause unforeseen side effects. Their poor zeta potential leads to aggregation, impacting delivery, triggering immune responses, and shortening circulation time. Determining the optimal dosage is challenging due to factors like delivery methods, short half-life, and variations in parent cell origins. Additionally, it remains uncertain whether exosomes or specific types, engineered versions, or alternatives like exosome-mimicking liposomes are the best therapeutic option [268]. Further research is essential to better understand how tumor-derived exosomes contribute to immune evasion, angiogenesis, metastasis, and therapy resistance. Specifically, exploring the role of exosomes in immune modulation, particularly in the regulation of immune checkpoints, and engineering exosomes for targeted drug delivery, especially for chemotherapy and gene therapy, could pave the way for novel immunotherapeutic strategies.

Inflammasomes play a dual role in cancer progression, both promoting and suppressing tumor growth by regulating inflammation and immune responses. While inflammasome-targeted therapies hold significant promise in cancer treatment, several challenges persist. The specificity of inflammasome inhibitors and activators is critical to prevent unintended disruptions in immune pathways. For instance, indiscriminate IL-1β blockade may impair host defenses against infections, whereas widespread inflammasome activation could trigger excessive inflammation and tissue damage. Additionally, the heterogeneity of the TME complicates the development of universal inflammasome-targeting strategies, as their roles vary across cell types. In TAMs and DCs, inflammasomes may enhance anti-tumor immunity, whereas in tumor cells, they can promote malignancy. A deeper understanding of inflammasome functions in specific TME cell subsets will be essential for designing precise and effective therapies.

Furthermore, while combining inflammasome-targeting agents with immune checkpoint inhibitors shows promise, further research is needed to determine the most effective combinations and assess their long-term impact on immune function and tumor progression. Ultimately, inflammasome-targeted therapies could become a powerful tool in cancer immunotherapy, but their success will hinge on enhancing selectivity, deciphering the intricate immune dynamics within tumors, and developing personalized treatment strategies.

Recent advancements in research techniques, particularly in bioinformatics, have provided powerful tools for elucidating inflammasome expression, function, and their associations with clinical outcomes. For instance, a pan-cancer study demonstrated that NLRP3 expression varies across tumor types, with its activity either elevated or suppressed depending on the cancer type [269]. Notably, this analysis highlighted a strong correlation between NLRP3 expression and patient survival, especially in melanoma and hepatocellular carcinoma, where higher NLRP3 levels were linked to improved survival rates, a more favorable prognosis, and enhanced responses to immunotherapy. Additionally, another study developed a risk-scoring system based on inflammasome-related genes to predict clinicopathologic features, prognosis, and immune response patterns in kidney renal clear cell carcinoma [270]. These findings underscore the need for further research to clarify the role of inflammasomes in cancer progression and their potential clinical applications.

Exosomes play a multifaceted role in regulating inflammasome activation within the TME. By carrying and transferring bioactive molecules, they serve as key mediators of inflammation and immune responses in cancer. Exosomes derived from immune cells, epithelial cells, and cancer cells have been shown to enhance inflammasome activation, triggering the release of pro-inflammatory cytokines such as IL-1β and IL-18. Moreover, the selective packaging of biomolecules into exosomes is highly regulated and varies based on the cell of origin, further shaping their impact on recipient cells. While growing evidence underscores the link between exosomes and inflammasome activation, several key questions remain about the underlying mechanisms of these interactions. Although inflammasome activation is known to promote exosome release, its precise impact on exosome secretion is not yet fully understood. Discrepancies in research findings may stem from differences in the types of PRRs involved, the specific activators used, and the targeted effector cells. To clarify these processes, further studies are needed to examine how these factors influence exosome–inflammasome dynamics and to explore the therapeutic potential of targeting these pathways in cancer treatment.

## Figures and Tables

**Figure 1 ijms-26-02716-f001:**
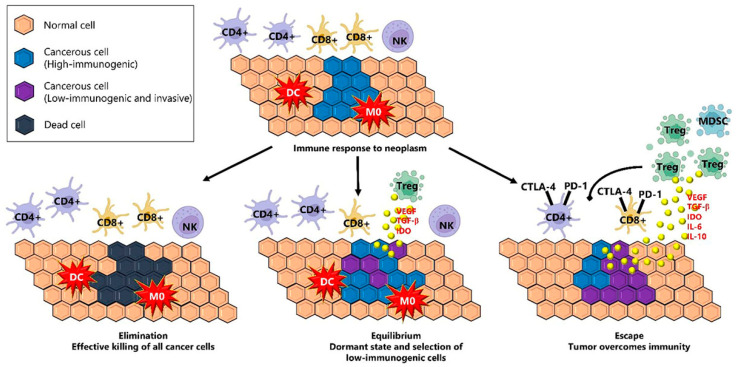
The immunoediting process includes three stages: elimination, equilibrium, and escape, highlighting different subtypes of immune cells involved in tumors. The figure is adapted from De Mello et al. [47].

**Figure 2 ijms-26-02716-f002:**
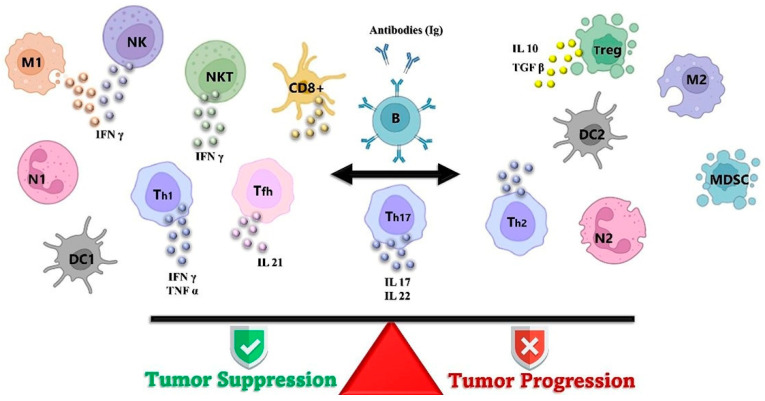
The intracellular interaction between different leukocyte subsets and their predominant roles in either stimulating or inhibiting tumor growth, including myeloid lineage leukocytes, tumor-associated macrophages (M) with both pro- and antitumor properties, various helper T-cell subsets (Th), cytotoxic T cells, regulatory T cells (Tregs), dendritic cells (DCs), natural killer cells (NK), neutrophils (N), B cells, and myeloid-derived suppressor cells (MDSC). The figure is adapted from Salgado et al. [51].

**Figure 4 ijms-26-02716-f004:**
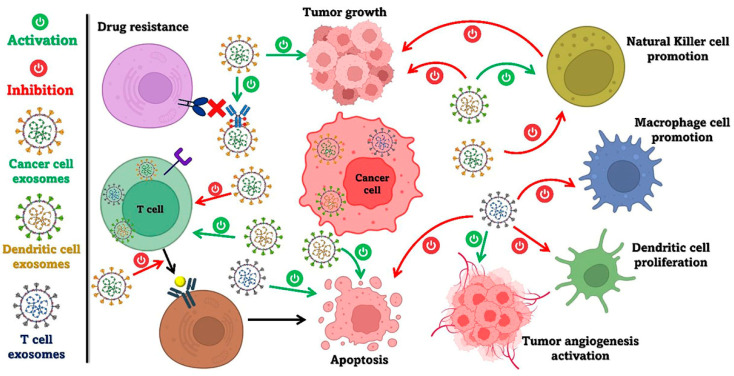
Origins of exosomes and their impact on tumor progression. The figure is adapted from Awadasseid et al. [116].

**Table 1 ijms-26-02716-t001:** Differences in the behavior of different immune cells.

Immune Cell	Normal Conditions (Antitumor) (Tumor Regression, Rejection, Apoptosis, Cytotoxic Good Prognosis)	Tumor Conditions (Protumor) (Tumor Growth, Spread, Metastasis, Poor Prognosis)	References
Macrophages	M1 can produce cytokines and establish an environment that enhances immune defense in reaction to inflammation.	M1 can release M1-Th1, which plays a defensive role against tumor cells. M2 macrophages in tumor conditions can release IL-10, angiogenic factors, and tumor growth factor β (TGF-β).	[27,28,29]
Fibroblasts	Fibroblasts have an important role in the healing process following inflammation or injuries.	Fibroblasts can transform into cancer-associated fibroblasts, which can regulate cytokines, myeloid suppressor cells, and Tregs, thus contributing to the progression and dissemination of tumors.	[30,31]
Endothelial cells	Endothelial cells have a role in inflammation, the process of regeneration, and healing through producing substances such as tumor necrosis factor α (TNFα) and other specific interleukins.	Endothelial cells exhibit an altered structure due to angiogenesis, resulting in impaired immune cell function.	[32]
Regulatory T cells (Tregs)	Tregs play a key role in regulating the immune system.	Tregs stimulate the release of IL-10 and TGF-β.	[33,34]
Neutrophils	Neutrophils participate in phagocytosis and generate cytokines.	Neutrophils can be categorized into two types: N1 and N2. N1 exerts an antitumoral effect by attracting IL-8 from tumor cells, and N2 activates a pro-tumoral effect by contributing to angiogenesis.	[35]
Eosinophils	Eosinophils, under normal circumstances, demonstrate antiparasitic actions and contribute to immune responses.	Eosinophils exhibit a tumoricidal role by releasing interleukins such as IL-2 and IL-4.	[36,37]
γδ T-cells	These T-cells can respond to phosphor antigens and communicate antigens to CD8+ and CD4+ lymphocytes, besides collaborating with natural killer (NK) cells.	γδ T-cells exhibit the strongest positive correlation with cancer prognosis. These cells also include IL-17-secreting cells, which can trigger the production of vascular endothelial growth factors and other angiogenesis-related factors.	[38,39]
Natural killer (NK) cells	NK cells are the primary antitumor defenders. They enhance the action of T-helper 1 lymphocytes (Th1) and stimulate CD8+ lymphocytes.	NK cells interacting with tumor cells usually express the two CD45 isoforms; CD45RA and CD45RO. The anti-tumor NK cells perform trogocytosis on tumor markers, facilitating their identification.	[40,41]
Dendritic cells (DCs)	DCs attract lymphocytes to antigen-presenting cells (APCs).	DCs draw lymphocytes to tumor-presenting cells, thus, their invasion of tumor cells is associated with delayed cancer progression and, in turn, a favorable prognosis.	[40,42]
Type 1 CD8+ T cells	These cells are the primary defense against cancer in humans.	They become activated when tumor antigens are presented by a dendritic cell along with attracting M1 macrophages, T-helper 1 lymphocytes (Th1), and T-helper 9 lymphocytes (Th9) to the tumor cells. Their presence indicates a favorable prognosis in tumor cases.	[43,44]
CD4+ T cells, Th1, Th2	CD4+ T lymphocytes display diverse polarization based on the specific cytokine combinations influencing them. The Th1 polarization is driven by the presence of IFN and IL-12 from M1 macrophages, as well as IL18, IL-27, and IL1, all of which are involved in the anti-tumor defense.	Th2 polarization is driven by specific interleukins (ILs) released by mast cells, NK cells, and CD4+ memory. The Th2 cells respond by producing other cytokines, such as interleukins (ILs)- 4, 5, 10, 13, 25, and 33, impairing the effectiveness of the immune system in malignancies.	[45,46]
B cells	B cells act as a pro-tumoral in some malignancies by releasing IFN-γ and IL-12	B cells stimulate IL-10 and TGF-β release, which have a role in tumor progression.	[34]

**Table 2 ijms-26-02716-t002:** Summary of the origins of the exosomes and their influence on tumorigenesis.

Origin of Exosomes	Mechanism and Effect Observed	References
Tumor cells	Exosomes can exhibit diverse functional roles, contingent upon their cellular origin and the surrounding environmental conditions. They activate cytotoxic T cell (CTL) responses specific to tumor antigens while simultaneously inhibiting leukocyte proliferation by downregulating ZAP70 and ERK1, 2. Also, they can impair natural NK cell cytotoxicity and hinder the proliferation of CD8+ and CD4+ T cells, potentially diminishing the immune system’s effectiveness in fighting cancer.	[117,120]
Dendritic cells (DCs)	Exosomes can trigger potent anti-cancer effects. Exosomes produced by DCs, containing chaperones like MHC I, MHC II, HSP70-90, and CD86, can stimulate CD4+ and CD8+ T cells. Exosomes derived from alpha-fetoprotein-expressing DCs induce the production of more IFN-expressing CD8+ T cells, triggering IFN and IL-2 levels, and reduce the levels of CD25+Foxp3+ Tregs, TGF, and IL-10.	[126,127,131]
B-lymphoma cell	Exosomes induce apoptosis in CD4+ T lymphocytes via MHC II. B lymphoma cells induced by heat shock released exosomes with elevated levels of HSP90 and HSP60, along with heightened immunogenicity molecules, and these exosomes effectively stimulate CD8+ T cells, yielding an anti-cancer effect. When DCs interact with exosomes from B cell lymphoma cells, they can enhance the activation of T cells, leading to the release of TNF-α and IL-6 while simultaneously decreasing IL-4 and IL-10 production.	[132,133,134]
T-lymphocytes	Exosomes derived from activated CD8+ T cells effectively hinder tumor invasion and metastasis mediated by fibroblastic stroma. Additionally, T cell exosomes express the CD63 protein and contain specific miRNAs that regulate immune responses and immune system development, playing a pivotal role in enhancing interaction between antigen-presenting T cells. Exosomes produced by CD4+ T cells can employ target cells through CD4–MHC interactions, ultimately leading to the elimination of immune-deficient cells	[138,139,142]
Natural killer (NK) cells	Exosomes are equipped with cytotoxic proteins such as FasL and perforin, besides characteristic NK markers like CD56. Additionally, NK-derived exosomes possess the ability to infiltrate tumor tissues directly, enabling them to exert their cytolytic effects.	[144,145]
Myeloid-derived suppressor cell (MDSC)	Exosomes derived from MDSCs carry cargo that matches their role in mediating immunosuppression. Additionally, MDSC-derived miR-126a+ exosomes have been shown to stimulate metastasis and confer resistance to therapy.	[149,150]
Tumor-associated macrophages (TAMs)	Exosomes originated from TAMs can create an immune-suppressive environment and enhance the progression of ovarian cancer by transferring miRNAs into CD4+ T cells. Moreover, M2 macrophage-derived exosomes transmit oncogenic miRNAs, promoting cancer cell invasion, migration, and resistance to chemotherapy.	[152,153]
Mast cells (MCs)	MC-derived exosomes promote cancer cell growth by transferring the KIT protein. Additionally, exosomes from MCs that express CD63 and OX40L result in the proliferation and development of CD4+ Th2 cells. Furthermore, MC-derived exosomes stimulate immature DCs to up-regulate molecules such as MHC II, CD40, CD80, and CD86, enabling T cells to present antigens and initiate the development of immune responses.	[156,157]
Neutrophils	Exosomes released by neutrophils adhere to and degrade the extracellular matrix through the actions of neutrophil elastase (NE) and the integrin Mac-1, facilitating the progression of inflammatory diseases. Furthermore, these exosomes inhibit the proliferation and migration of endothelial cells, thereby hindering pathological angiogenesis.	[158,159]

## Data Availability

Not applicable.

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
