# Peer review of "The Role of the Tumor Microenvironment (TME) in Advancing Cancer Therapies: Immune System Interactions, Tumor-Infiltrating Lymphocytes (TILs), and the Role of Exosomes and Inflammasomes"

_ijms, 2025, doi:10.3390/ijms26062716_

Round 1
Reviewer 1 Report
Comments and Suggestions for Authors
In this study entitled "The Role of Tumor Microenvironment (TME) in Advancing Cancer Therapies: Immune System Interactions, Tumor-infiltrating Lymphocytes (TILs), and the Role of Exosomes and Inflammasomes" the authors discussed the intricate role of the Tumor Microenvironment (TME) in advancing cancer therapies, highlighting the complex interplay between immune system components, tumor-infiltrating lymphocytes (TILs), exosomes, and inflammasomes.
While various components of TME were listed, there is a lack of critical summary and review of these components. The Conclusion, Challenges, and Future Directions section is somewhat vague and lacks depth. In addition, the following issues should be addressed:
- The introduction mentions that tumor cells evade the immune system through various mechanisms, The introduction should briefly present the impact of tumor microenvironment heterogeneity on immunotherapy.
- It is recommended to include a table in 2. Differences in cell behavior between normal and cancerous conditions summarizing the differences in the behavior of different immune cells under normal and tumor conditions and their impact on tumor progression.
- The section 5. History of tumor-infiltrating lymphocytes (TILs), while informative, is somewhat irrelevant to the topic of this article, which focuses on tumor microenvironment (TME) and immune system interactions in cancer therapy.
- Could the authors provide further clarification regarding the availability of relevant clinical trial data on Tumor-Infiltrating Lymphocyte (TIL) therapy, particularly with respect to its lymphodepletion regimen? It would be beneficial if the manuscript could incorporate these clinical findings.
- In 11. Exosomes origin can indicate their Influence on tumorigenesis section, the authors provide an extensive list of exosomes derived from various sources. It's recommended to summarize these in a table format to improve readability.
Author Response
We thank to the reviewers for their valuable comments and suggestions. The comments have been addressed and highlighted in red in the revised version. Additionally, a point-by-point response have been provided in the attached word document.

Reviewer 2 Report
Comments and Suggestions for Authors
The article presents the role of tumor microenvironment, exosomes and inflammasomes in cancer therapies. The article describes in detail the behavior of the cells within the tumor and what role TIL, exosomes and inflammasomes play within the tumor microenvironment.
- Nevertheless, the article lacks coherence. For example:
- The names of the cytokines vary throughout the article: one time it's IL-2, another time it's IL2.
- Titles of sections and subsections: in one case all words are capitalized, in another only a few words or only the first word.
- Use of abbreviation: only as the first time abbreviation are used an explanation is needed.
- I also have a question about the figures. Do the authors have permission from the authors to use these images as they look?
- There are several errors in the text:
- Lines 36-42: abbreviations should be explained and use a comma instead of semicolon
- Line 123: …On the other hand, Eosinophils under…
- Line 152: … and CD4+ memory (CD4+ memory)…
- Line 331: … can indicate their Influence on tumorigenesis…
- Line 375: … higher levels of IFN- and…
Author Response

(The authors gave the same response as above.)

Round 2
Reviewer 1 Report
Comments and Suggestions for Authors
I have carefully examined the authors' responses to my previous comments and the corresponding revisions made to the paper. Their point-by-point responses are satisfactory, the newly added tables improve the organization and clarity of this review.
The current version adequately addresses the deficiencies identified in the previous manuscript.